# The Herbal Combination CPA4-1 Inhibits Changes in Retinal Capillaries and Reduction of Retinal Occludin in db/db Mice

**DOI:** 10.3390/antiox9070627

**Published:** 2020-07-16

**Authors:** Young Sook Kim, Junghyun Kim, Chan-Sik Kim, Ik Soo Lee, Kyuhyung Jo, Dong Ho Jung, Yun Mi Lee, Jin Sook Kim

**Affiliations:** 1Research Infrastructure Team, Herbal Medicine Division, Korea Institute of Oriental Medicine, Daejeon 34054, Korea; ykim@kiom.re.kr (Y.S.K.); knifer48@kiom.re.kr (I.S.L.); jopd7414@kiom.re.kr (K.J.); 2Herbal Medicine Research Division, Korea Institute of Oriental Medicine, Daejeon 34054, Korea; dvmhyun@jbnu.ac.kr (J.K.); jdh9636@kiom.re.kr (D.H.J.); candykomg@kiom.re.kr (Y.M.L.); 3Department of Oral pathology, School of Dentistry, Chonbuk National University, Jeonju 54896, Korea; 4Clinical Medicine Division, Korea Institute of Oriental Medicine, Daejeon 34054, Korea; chskim@kiom.re.kr

**Keywords:** diabetic retinopathy, db/db mice, Cinnamomi Ramulus, Paeoniae Radix, CPA4-1, blood-retinal barrier, occludin

## Abstract

Increased formation of advanced glycation end products (AGEs) plays an important role in the development of diabetic retinopathy (DR) via blood-retinal barrier (BRB) dysfunction, and reduction of AGEs has been suggested as a therapeutic target for DR. In this study, we examined whether CPA4-1, a herbal combination of Cinnamomi Ramulus and Paeoniae Radix, inhibits AGE formation. CPA4-1 and fenofibrate were tested to ameliorate changes in retinal capillaries and retinal occludin expression in db/db mice, a mouse model of obesity-induced type 2 diabetes. CPA4-1 (100 mg/kg) or fenofibrate (100 mg/kg) were orally administered once a day for 12 weeks. CPA4-1 (the half maximal inhibitory concentration, IC_50_ = 6.84 ± 0.08 μg/mL) showed approximately 11.44-fold higher inhibitory effect on AGE formation than that of aminoguanidine (AG, the inhibitor of AGEs, IC_50_ = 78.28 ± 4.24 μg/mL), as well as breaking effect on AGE-bovine serum albumin crosslinking with collagen (IC_50_ = 1.30 ± 0.37 μg/mL). CPA4-1 treatment ameliorated BRB leakage and tended to increase retinal occludin expression in db/db mice. CPA4-1 or fenofibrate treatment significantly reduced retinal acellular capillary formation in db/db mice. These findings suggested the potential of CPA4-1 as a therapeutic supplement for protection against retinal vascular permeability diseases.

## 1. Introduction

Hyperglycemia induces the formation and accumulation of advanced glycation end products (AGEs), and these products are present at high levels in the blood and tissue of diabetic patients [1,2]. AGEs are accumulated at high levels in the tissues of patients with age-related diseases, such as chronic obstructive pulmonary disease, cardiovascular diseases, osteoporosis, and neurodegenerative diseases [3]. AGEs are formed by oxidative and non-oxidative reactions, and they affect the biochemical and physical properties of proteins in tissues. AGE formation is triggered by high glucose-induced oxidative stress and fluorescent protein cross-linking [4].

Diabetic retinopathy (DR) is a complication of diabetes that causes damage to retinal blood vessels [5]. Elevated AGE levels increase the breakdown of the blood-retinal barrier (BRB), adhesion of leukocytes, and retinal vascular injury, leading to serious impairment of vision. The BRB consists of inner and outer nuclear layers. The inner nuclear layer of the BRB consists of tight junctions between endothelial cells and pericytes, whereas the outer nuclear layer of the BRB is formed by tight junctions between retinal pigment epithelial cells [6]. The advent of anti-vascular endothelial growth factor (VEGF) has shown a remarkable effect in DR patients; however, most of DR patients have failed to achieve significant clinical visual improvement. The treatment of DR remains challenging. Inhibition of AGE formation has been suggested as a therapeutic target for improving insulin resistance in diabetes with obesity [7]. For example, fenofibrate, a peroxisome proliferator-activated receptor alpha (PPARα) agonist, was approved for slowing down the progression of DR in patients with type 2 diabetes mellitus in October 2013 in Australia [8]. Moreover, pyridoxamine, an inhibitor of AGE formation, has been shown to ameliorate insulin resistance in obese, type 2 diabetic mice [9]. The identification of inhibitors of AGE formation from natural sources has gained much attention.

During the last 15 years, we have screened inhibitors of AGE formation from natural products [10,11,12]. *Aster koraiensis* extract prevents retinal pericyte apoptosis in streptozotocin (STZ)-induced diabetic rats [13]. *Osteomeles schweinae* extract inhibits methylglyoxal (an active precursor in the formation of AGEs)-induced apoptosis in human retinal pigment epithelial cells [14]. Cinnamomi Ramulus (the twig of *Cinnamomum cassia* Blume; Lauraceae) and Paeoniae Radix (the root of *Paeonia lactiflora* Pallas; Paeoniaceae) have been shown to exert efficacy in inhibiting the formation of AGEs in our previous study. Cinnamomi Ramulus has traditionally been used for its anti-inflammatory, antioxidant, and neuroinflammatory effects [15]. Its marker compounds include coumarin, cinnamyl alcohol, and cinnamic acid. In humans, the effect of cinnamon is controversial; it significantly decreases plasma glucose to the baseline levels, without causing adverse effects nor significant glycemic and inflammatory indicators in patients with type 2 diabetes [16,17]. Paeoniae Radix has been used in traditional medicine for treating inflammatory diseases owing to its anti-allergic, immunoregulatory, and analgesic effects [18]. The marker compounds of Paeoniae Radix include gallic acid, albiflorin, paeoniflorin, and benzoic acid [19]. In a preliminary study, we evaluated the efficacy of inhibition of AGE formation with different combinations of the two herbs to obtain the best formulation. It showed a different inhibitory effect according to the ratio, and it was the best at CPA 4-1 (Cinnamomi Ramulus:Paeoniae Radix = 1:8). Here, we tested a mixture of the CPA4-1 to investigate the optimum ratio for inhibiting AGE formation in the human retinal pigment epithelial cells (ARPE-19). In addition, we examined the therapeutic efficacy of CPA4-1 in preventing DR in db/db mice, a well-established model of obesity-induced type 2 diabetes with retinal neurodegeneration [20,21].

## 2. Materials and Methods

### 2.1. Preparation of the CPA4-1

Cinnamomi Ramulus and Paeoniae Radix were purchased from a traditional herbal medicine store in Daejeon, Republic of Korea, in April 2016 and identified by Prof. Ki Hwan Bae (College of Pharmacy, Chungnam National University, Republic of Korea). Voucher specimens of Cinnamomi Ramulus (KIOM-CIRA-2016) and Paeoniae Radix (KIOM-PARA-2016) have been deposited in the Herbarium of Korea Institute of Oriental Medicine (KIOM), Republic of Korea. The herbal combination was prepared at a Cinnamomi Ramulus to Paeoniae Radix ratio of 1:8 (*m*/*m*). For preparing CPA4-1 extract, 20 g of Cinnamomi Ramulus and 160 g of Paeoniae Radix were weighed accurately and mixed. Distilled water (1080 mL) was added to the mixed herbs (180 g) and extracted at 100 °C for 3 h using a reflux extractor (MS-DM607, M-TOPS, Seoul, Korea). The extract solutions were filtered and evaporated under reduced pressure using a rotary evaporator (N-1200A; Eyela, Tokyo, Japan) at 50 °C and then freeze-dried using a freeze dryer (FDU-2100; Eyela) at −80 °C for 72 h to obtain an extract powder of CPA4-1 (18.5 g; yield, 10.3%). This sample extract (100 mg) was dissolved in 50% methanol (10 mL), and the solution was filtered through a 0.45-μm syringe filter (Whatman, Clifton, NJ) prior to injection. Standard stock solutions of five reference standards (all at 1 mg/mL) were prepared in HPLC-grade MeOH, stored at <4 °C, and used for HPLC analyses after serial dilution in MeOH.

### 2.2. HPLC Analysis

HPLC analyses were performed using an Agilent 1200 HPLC instrument (Agilent Technologies, Santa Clara, CA, USA) equipped with a binary pump (G1312A), vacuum degasser (G1322A), auto-sampler (G1329A), column compartment (G1316A), and diode array detector (DAD, 1365B). Data were collected and analyzed using the Agilent ChemStation software. Chromatographic separation was conducted using a Luna C18(2) (250 × 4.6 mm, 5.0 μm; Phenomenex, Torrance, CA, USA), and the column temperature was maintained at 40 °C. The mobile phase consisted of 0.1% formic acid in water (A) and acetonitrile (B) with gradient elution for better separation. The gradient solvent system was optimized as follows: 95–55% A (0–40 min), 55–0% A (40–41 min), 100% B (41–45 min), and 95% A (45–55 min). The flow rate was 1 mL/min. The detection was conducted at 240 nm, and the injection volume of each sample was 10 μL. To test for linearity, standard solutions at five levels were prepared by serially diluting the stock solution. Each analysis was repeated three times, and the calibration curves were fitted by linear regression. The limit of detection (LOD) and limit of quantification (LOQ) data obtained under the optimal chromatographic conditions were determined using signal-to-noise (S/N) ratios of 3 and 10, respectively.

### 2.3. Determination of Preventive Effect of CPA4-1 on AGE Formation

Bovine serum albumin (BSA) (Sigma-Aldrich, St. Louis, MO, USA) was mixed with 0.2 M glucose or fructose in 50 μM phosphate buffer at 10 mg/mL. Various concentrations of CPA4-1 extract or aminoguanidine (AG) (Sigma-Aldrich) were added, and the mixture was incubated at 37 °C for 14 days. After incubation, the fluorescence products of glycated BSA was determined using a spectrofluorometer (Synergy HT; BIO-TEK, Winooski, VT, USA) at excitation/emission wavelength of 350/450 nm. The 50% inhibition concentration (IC_50_) for AGE formation was calculated via interpolation from the concentration-inhibition curve.

### 2.4. Breaking Effect of CPA4-1 on Preformed AGE-Collagen Complexes

The ability of CPA4-1 to break preformed AGEs was evaluated using a previously described method [22]. Briefly, 1 μg of glycated BSA (AGEs-BSA) (MBL International, Woburn, MA, USA) was pre-incubated in collagen-coated 96-well plates for 24 h, and the collagen-AGEs-BSA complexes were then incubated with CPA4-1. Collagen-AGE-BSA crosslinking was detected using mouse anti-AGEs primary antibody (Clone No. 6D12; Trans Genic Inc., Kobe, Japan), horseradish peroxidase-linked goat anti-mouse IgG secondary antibody, and H_2_O_2_ substrate containing 2,2′-azino-bis (3-ethylbenzothiazoline-6-sulfonic acid) (ABTS) chromogen. Breakdown levels were measured as the percentage of decrease in optical density (OD = 410 nm). We calculated the IC_50_ (μg/mL) as 50% inhibition of collagen-AGE-BSA crosslinking, in which the crosslinking inhibition percent was calculated as follows:

Inhibition of collagen-AGE-BSA crosslinking (%) = $100-\frac{\mathrm{abserbance}\;\mathrm{of}\;\mathrm{sample}}{\mathrm{abserbance}\;\mathrm{of}\;\mathrm{control}\;}\times 100$.

Breaking of AGE-induced crosslinking was expressed as the percentage decrease in optical density. The breaking percentage was calculated according to the above equation.

### 2.5. Cell Culture and Determination of Preventive Effect of CPA4-1 on AGE Formation in ARPE-19 Cells

ARPE-19 cells were purchased from the American Type Culture Collection (ATCC CRL-2302; Manassas, VA, USA) and maintained at 37 °C in a humidified 5% CO_2_ incubator [23]. To examine the inhibitory effect of CPA4-1 on AGE formation, the cells were treated with CPA4-1 (10, 20, or 50 μg/mL in DMSO) for 1 h before the addition of 25 mM glucose and 500 μg/mL BSA (Roche Diagnostics, Basel, Swiss). After that, the cells were incubated for 48 h and then subjected to Western blotting analysis.

### 2.6. Western Blot Analysis

Sodium dodecyl sulfate-polyacrylamide gel electrophoresis (SDS-PAGE) was performed, as described previously [4]. Each sample (25 μg/mL) was fractionated on a 10% SDS-PAGE, after which the proteins were transferred to a polyvinylamide gel membrane (Millipore, Billerica, MA, USA) using traditional tank transfer system (Mini Trans-Blot cell, Bio-rad, Hercules, CA, USA). Membranes were probed with antibodies against AGEs (Clone No. 6D12; Trans Genic Inc.), occludin (Invitrogen, Carlsbad, CA, USA), and β-actin (Sigma-Aldrich), each at 1:1000 dilution. Signals were detected using a WEST-one ECL solution (Intron, Korea) and captured on a Fuji Film LAS-3000 (Tokyo, Japan).

### 2.7. Animals and Experimental Design

Animal experiments were conducted by Qu-Best Bio Co., Ltd. (Yongin City, Korea), according to the National Institutes of Health Guide for the Care and Use of Laboratory Animals (Approval number: QJE14015). Male C57BL/KsJ db/db mice and their age-matched lean littermates (db/+) were purchased from Japan SLC (Shizuoka, Japan). At 8 weeks of age, the db/db mice were randomly assigned into four groups (n = 10): NOR, normal control mice; DM, db/db mice; FENO, db/db mice treated with fenofibrate (100 mg/kg); CPA4-1-100, db/db mice treated with CPA4-1 (100 mg/kg). CPA4-1 was dissolved in the vehicle (0.5% *w*/*v* carboxymethyl cellulose solution) at a concentration of 5 mg/mL. The mice received daily gastric gavage of fenofibrate (100 mg/kg) or CPA4-1-100 (100 mg/kg), and db/+ mice received the same vehicle treatment for 12 weeks. Blood glucose level was measured with an automated biochemistry analyzer (HITACHI917; Hitachi, Japan), and the glycated hemoglobin (Hb1Ac) level was determined by a commercial kit (Roche Diagnostic, Mannheim, Germany).

### 2.8. Measurement of BRB Permeability

At autopsy, mice were anesthetized by intraperitoneal injection of 10 mg/kg zolazepam (Zoletil, Virbac, Carros, France) and 10 mg/kg xylazine hydrochloride (Rumpun, Bayer, Frankfurt, Germany). The peritoneal and thoracic cavities were opened to secure the heart, and 50 mg/mL fluorescein-dextran (10 kDa Mw, Sigma-Aldrich) and 10 mg/mL Hoechst 33342 (Sigma-Aldrich) dissolved in 1 mL sterile phosphate-buffered saline (PBS) were injected into the left ventricle. After 5 min, the eyeballs were removed, fixed in 4% paraformaldehyde for 2 h, and the retina was separated from the eyecup. The separated retina was placed on a slide, mounted with an aqueous mounting medium, and observed under a fluorescence microscope with digital capture (BX41 microscope; Olympus, Tokyo, Japan).

### 2.9. Preparation of Trypsin-Digested Retinal Vessel

The isolated retinas were placed in 10% formalin for 2 days. After fixation, the retina was incubated in trypsin (3% in sodium phosphate buffer containing 0.1 M sodium fluoride) for 60 min. The vessel structures were separated from retinal cells by gentle rinsing in distilled water. The vascular specimens were mounted on a slide and subjected to periodic acid-Schiff staining. The specimens were then analyzed under a microscope with digital capture (BX41 microscope; Olympus). The number of acellular capillaries per mm^2^ of the capillary area was determined by counting 10 selected microscopic fields.

### 2.10. Immunohistochemical Staining

Each eye was enucleated and fixed with 4% paraformaldehyde for 24 h. The retina was isolated under a dissecting microscope and entirely washed with water and incubated in 3% trypsin in sodium phosphate buffer for 1 h. The trypsin digests were subjected to immunofluorescence staining, as previously described [13]. The slides were incubated with a mouse anti-occludin antibody (Invitrogen) for 1 h at room temperature. Signals were detected using a rhodamine-conjugated goat anti-mouse antibody (Santa Cruz Biotechnology, Paso Robles, CA, USA) and detected by fluorescence microscopy (BX51; Olympus).

### 2.11. Statistical Analysis

Data were expressed as mean ± SD or mean ± SE. ANOVA with Tukey’s test was used for multiple comparisons using the Prism 7.0 software (GraphPad software, San Diego, CA, USA).

## 3. Results

### 3.1. HPLC Analysis of CPA4-1

The regression equations for the five reference standards, together with the LOD and LOQ values, are shown in Table 1. All of the calibration curves showed good linearity (R^2^ > 0.999) within the test ranges. An established analytical HPLC method was applied to quantitatively analyze five compounds in CPA4-1. The main chromatographic peaks in CPA4-1 were identified by comparing their retention times and UV spectra with those of the corresponding commercial standards. Representative chromatograms of CPA4-1 and five standard compounds monitored at a wavelength of 240 nm are shown in Figure 1. The content of the five compounds, namely, gallic acid, albiflorin, paeoniflorin, benzoic acid, and cinnamic acid, in CPA4-1 were 1.1%, 5.3%, 8.5%, 0.4%, and 0.1%, respectively (Table 2).

### 3.2. Inhibitory Effects of CPA4-1 on AGE Formation In Vitro and in ARPE-19 Cells

As shown in Table 3, CPA4-1 inhibited AGE formation compared with the positive control, AG. The IC_50_ value of CPA4-1 was 6.84 ± 0.08 μg/mL, and that of AG was 78.28 ± 4.24 μg/mL. CPA4-1 was approximately 11.44-fold more potent than AG in inhibiting AGE formation.

AGE-specific fluorescence was detected after incubation of AGEs-BSA with CPA4-1. The results showed that CPA4-1 was inhibited in a dose-dependent manner (Figure 2A). As shown in Figure 2B,C, CPA4-1 inhibited AGE formation in ARPE-19 cells in a dose-dependent manner.

### 3.3. Breaking Effect of CPA4-1 on Preformed AGE-Collagen Complexes

The ability of CPA4-1 to break the cross-links in the preformed AGE-collagen complexes was tested (Figure 3). CPA4-1 destroyed the cross-links in the complexes in a dose-dependent manner (IC_50_ = 1.30 ± 0.37 μg/mL).

### 3.4. Metabolic and Physical Parameters

Blood glucose and HbA1c levels are summarized in Table 4. Blood glucose and HbA1c levels significantly increased in the DM group (*p* < 0.05), but decreased in the FENO group. However, in the CPA4-1 group, glucose level was not changed, and HbA1c was slightly reduced, although not significant (Table 4).

As shown in Figure 4, compared with the normal mice group, the DM mice group exhibited significantly elevated body weight. However, treatment with CPA4-1 or FENO did not lead to any significant changes in body weight compared with that of DM mice.

### 3.5. Preventive Effects of CPA4-1 on BRB Breakage in db/db Mice

Fluorescence leakage due to BRB damage was observed and analyzed. In the normal group, no fluorescence leakage was observed. However, in the db/db mice group (DM), retinal vessels showed bright fluorescence due to the BRB breakage. As shown in Figure 5, the DM group showed a significant increase in BRB leakage compared with the normal group. FENO- or CPA4-1-treated db/db mice groups showed a significant decrease in BRB leakage compared with the db/db mice group.

### 3.6. Preventive Effects of CPA4-1 on Tight Junction Protein Loss in db/db Mice

To examine the therapeutic efficacy of CPA4-1 on the BRB, we detected loss of tight junction proteins in the retinal vascular endothelial cells of db/db mice. These proteins are important factors that increase the permeability of blood vessels. As shown in Figure 6A, the damage of vessels significantly increased in the retinas of db/db mice compared with that of the normal group. To elucidate whether this phenomenon was related to the expression of tight junction proteins, we assessed occludin levels in the retina of db/db mice. Figure 6B shows that the expression of occludin was downregulated in db/db mice compared with that in the normal control group. However, CPA4-1 treatment restored occludin expression in db/db mice.

### 3.7. Effects of CPA4-1 on Acellular Capillary Formation in db/db Mice

The formation of the acellular capillary, due to the loss of vascular endothelial cells and pericytes among blood vessels, occurred in DR. As shown in Figure 7, the db/db mice group showed a significant increase in retinal acellular capillary formation (black arrow) compared with the normal group. As expected, the acellular capillary of FENO- or CPA4-1-treated db/db mice significantly decreased compared with that of vehicle-treated db/db mice.

## 4. Discussion

In our previous studies, we examined the effects of herbal extracts in inhibiting AGE formation (CPA1-1 (1:1) IC_50_ = 9.94 ± 0.20 μg/mL; CPA2-1 (1:2) IC_50_ = 7.54 ± 0.18 μg/mL; CPA3-1 (1:4) IC_50_ = 8.16 ± 0.23 μg/mL; CPA4-1 (1:8) IC_50_ = 6.84 ± 0.08 μg/mL) and we showed potential therapeutic targets of DR in animal and in vitro models. In this study, we investigated whether CPA4-1, a herbal combination, inhibits AGE formation and breaks the cross-linking of AGE with collagen in vitro. We tested whether CPA4-1 affects the expression of occludin in the retina and whether this effect is correlated with inhibition of BRB breakage. Our findings showed that CPA4-1 significantly prevented retinal vascular leakage and retinal acellular capillary formation. These results indicated that CPA4-1 attenuated the downregulation of tight junction protein occludin in the retinas of db/db mice. The tight junction acts as a semipermeable barrier for paracellular transports of solutes, ions, water, and cells. Occludin is the first identified component of the tight junction strand, a transmembrane protein of approximately 60 kDa [24]. In diabetic patients, retinal occludin protein expression is selectively decreased, and BRB permeability is increased [25]. Decreased expression of tight junction proteins is important in diabetic patients, as it causes increased vascular permeability, vasodilation, and edema. As CPA4-1 was shown to increase the expression of occludin in the retina of db/db mice in this study, CPA4-1 might serve as an inhibitor of AGE formation to protect against DR.

Fasting blood glucose and HbA1c levels are important factors in diabetes. HbA1c is monitored in the long-term follow-up of patients with DR, and diabetic patients are primarily treated with insulin to lower blood glucose levels. Fenofibrate decreased blood glucose level and HbA1c level in the db/db mice group (Table 3). Fenofibrate treatment induces lowing of blood glucose levels and HbA1c by reducing adiposity, thus improving peripheral insulin action [26]. Previous studies have suggested that that fenofibrate exerts robust protective effects against DR in type 2 diabetic patients and type 1 diabetic animal models [27,28]. The potential therapeutic mechanisms of fenofibrate on retinal microvascular dysfunctions induced by diabetes include the restoration of VEGF and the reduction of oxidative stress in diabetic rats [29]. As CPA4-1 did not alter blood glucose level and HbA1c of db/db mice compared with those of the db/db mice group, CPA4-1 might act as a supplement for the prevention of BRB breakage and retinal acellular capillary formation, without altering blood glucose level or HbA1c of diabetic patients.

Herbal medicine has been used to treat diabetes and diabetic complications, including DR, for thousands of years worldwide. Many researchers have tried to provide evidence of the therapeutic effectiveness and mechanisms of herbal medicine. *Panax notoginseng* (Araliaceae) has been reported to exert antidiabetic activities, and it contains saponins, such as ginsenoside Re 14, ginsenoside Rd, ginsenoside Rg1, ginsenoside Rb1, and notoginsenoside R1. These saponins have also been reported to exert antioxidant action [30]. As free radical damage is critically involved in the pathophysiology of DR, *P. notoginseng* saponins can be used in the treatment of DR owing to their beneficial effects. *Pueraria lobata* (Fabaceae) is one of the most used herbs in traditional Korean medicine, and it contains puerarin, genistein, and daidzein. Puerarin directly inhibits the formation of AGEs in vitro [31]. CPA4-1 is formulated at the optimum conditions to inhibit AGE formation. Cinnamic acid from Cinnamomi Ramulus promotes the management of diabetes and its complications [32]. Cinnamic acid and its derivatives stimulate insulin and hepatic gluconeogenesis, enhance glucose uptake, and inhibit AGE formation [33,34]. Paeoniae Radix extract contains 5% paeoniflorin, the most important component that suppresses high glucose-induced retinal microglial matrix metalloproteinase 9 (MMP-9) expression and inflammation by reducing capillary permeability [35]. Paeoniflorin also exerts a neuroprotective effect on inflammation and apoptosis in Alzheimer’s disease and regulates hepatic cholesterol synthesis and metabolism by reducing oxidative stress [36,37].

## 5. Conclusions

The present study showed that CPA4-1 exerted an inhibitory effect on AGE formation and breaking effect on AGE-collagen cross-linking. CPA4-1 treatment ameliorated BRB leakage and retinal acellular capillary formation in db/db mice. These results suggested the potential of CPA4-1 as a therapeutic supplement for protection against retinal vascular permeability diseases.

## 6. Patents

The patent related to this study has been registered in Korea (no. 10-2015-0176274).

## Figures and Tables

**Figure 1 antioxidants-09-00627-f001:**
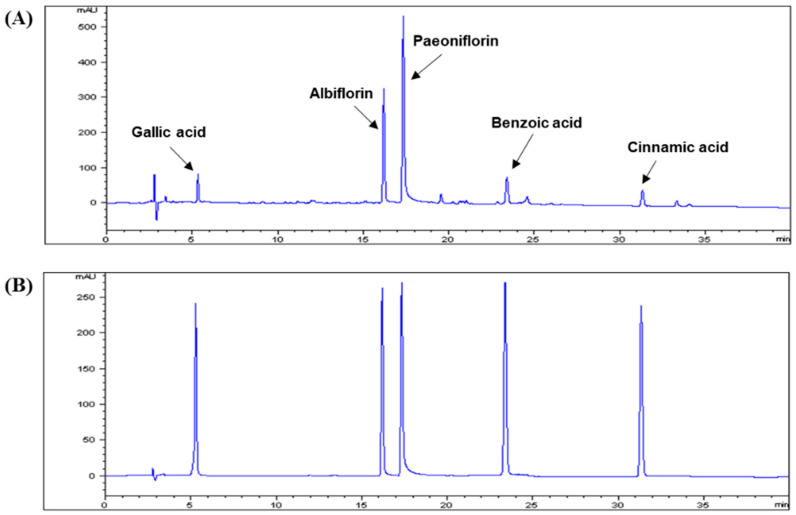
HPLC chromatograms of CPA4-1 (Cinnamomi Ramulus:Paeoniae Radix = 1:8) (**A**) and five standard compound mixtures (**B**). The chromatographic conditions are described in the text. Detection was conducted at 240 nm.

**Figure 2 antioxidants-09-00627-f002:**
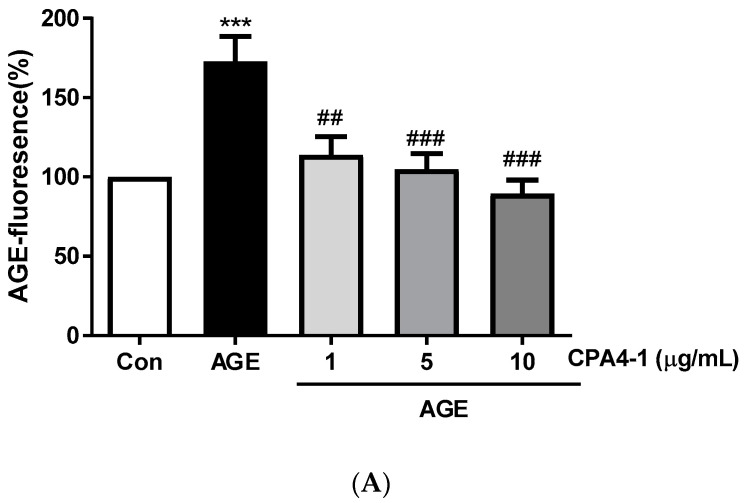
Inhibitory effect of CPA4-1 on the formation of advanced glycation end product (AGE). (**A**) AGE-fluorescence. *** *p* < 0.001 vs. Con; ^##^
*p* < 0.01, ^###^
*p* < 0.001 vs. AGE (n = 8). (**B**,**C**) Inhibitory effect of CPA4-1 on AGE formation in ARPE-19 cells. Data are expressed as mean ± SD. * *p* < 0.05 vs. Con; ^##^
*p* < 0.01, ^#^
*p* < 0.05 vs. High glucose (HG) + BSA (n = 4).

**Figure 3 antioxidants-09-00627-f003:**
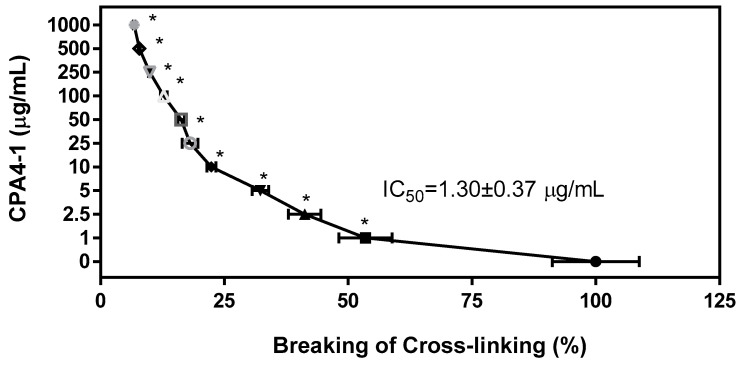
Breaking effect of CPA4-1 on AGE-BSA cross-linking with collagen. IC_50_ (50% inhibition concentration) values were calculated from the dose-inhibition curve. Data are expressed as mean ± SD (n = 4). * *p* < 0.001 vs. Con (0 μg/mL).

**Figure 4 antioxidants-09-00627-f004:**
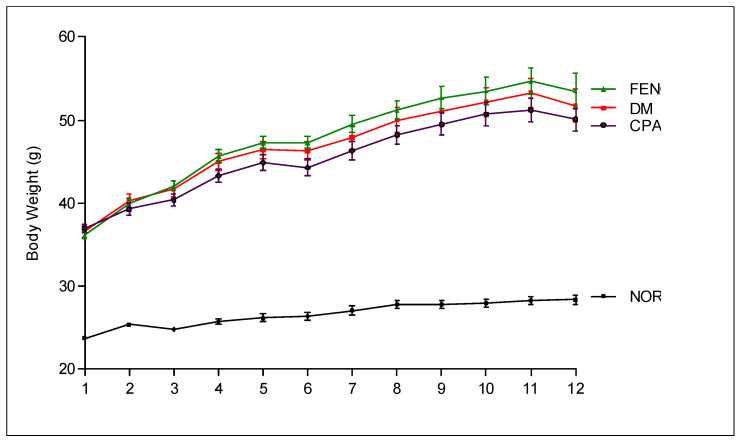
Body weight. NOR, normal control mice; DM, db/db mice; FENO, db/db mice treated with fenofibrate (100 mg/kg); CPA4-1, db/db mice treated with CPA4-1-100 (100 mg/kg). All data are expressed as mean ± SE.

**Figure 5 antioxidants-09-00627-f005:**
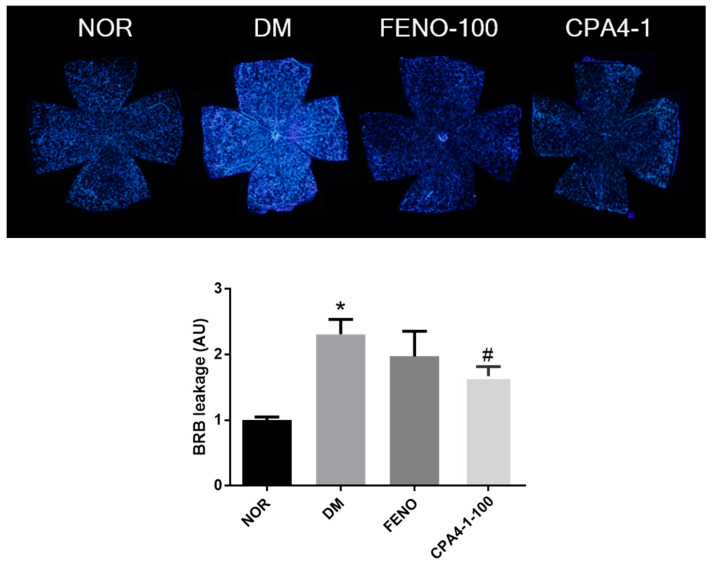
Effects of FENO and CPA4-1 on blood-retinal barrier (BRB) breakage in db/db mice. NOR, normal control mice; DM, db/db mice; FENO, db/db mice treated with fenofibrate (100 mg/kg); CPA4-1, db/db mice treated with CPA4-1-100 (100 mg/kg). All data are expressed as mean ± SE. * *p* < 0.05 vs. NOR group. ^#^
*p* < 0.05 vs. DM group.

**Figure 6 antioxidants-09-00627-f006:**
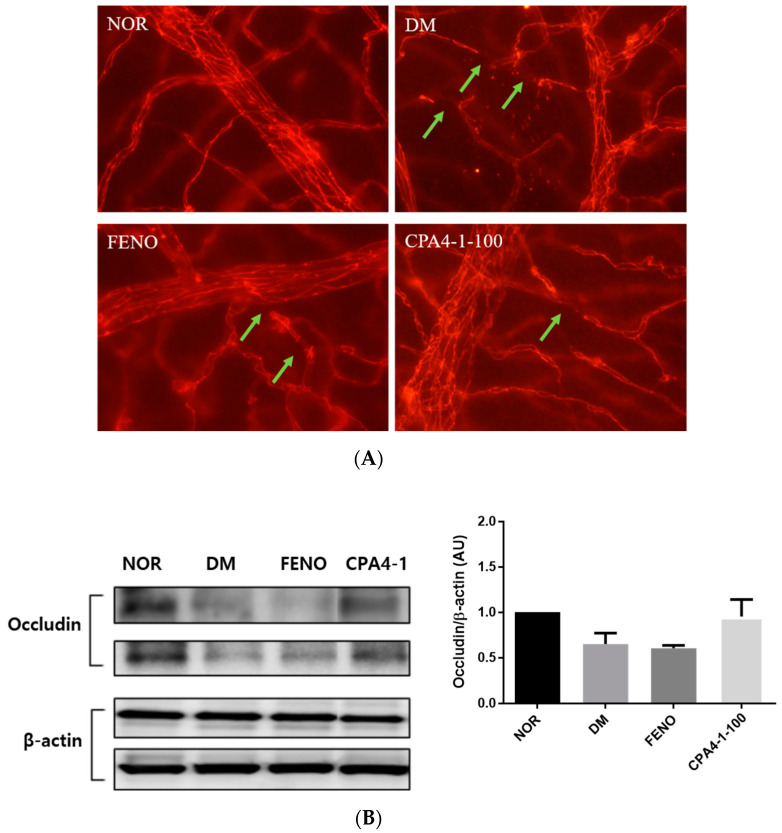
Effect of CPA4-1 on tight junction protein loss. (**A**) Immunofluorescence staining for occludin in trypsin-digested retinal vessels acellular capillaries (green arrow). (**B**) Expression of retinal occludin. The total protein was isolated from retinal tissues and subjected to Western blotting using occludin and beta-actin antibodies. Values in the bar graphs represent the mean (n = 2).

**Figure 7 antioxidants-09-00627-f007:**
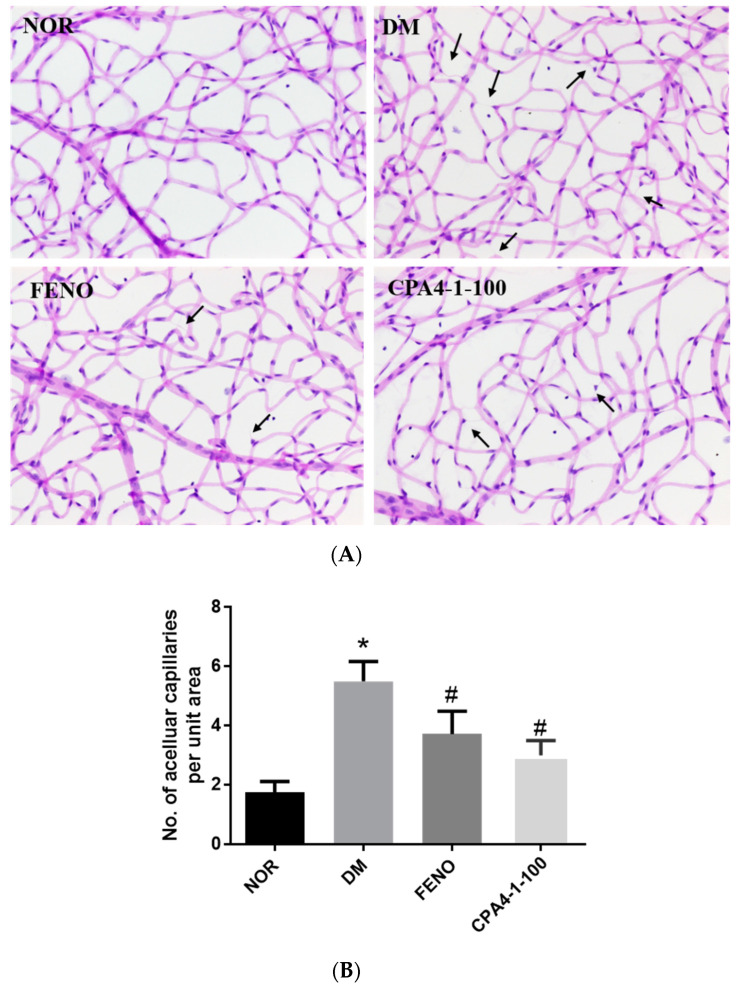
Effects of FENO and CPA4-1 on retinal acellular capillary formation in db/db mice. Retinal vessels were stained with H&E (**A**). Acellular capillaries (black arrow) were measured to assess the extent of retinopathy (**B**). NOR, normal control mice; DM, db/db mice; FENO, db/db mice treated with fenofibrate (100 mg/kg); CPA4-1, db/db mice treated with CPA4-1-100 (100 mg/kg). All data are expressed as mean ± SE (n = 6 ~ 7). * *p* < 0.05 vs. NOR group. ^#^
*p* < 0.05 vs. DM group.

**Table 1 antioxidants-09-00627-t001:** Regression equation, linearity, LOD, and LOQ for five marker compounds (n = 3).

Compound	Regression Equation ^a^	Linear Range (μg/mL)	Linearity (R^2^)	LOD ^b^ (μg/mL)	LOQ ^c^ (μg/mL)
Gallic acid	y = 11.507x − 8.142	6.25–200	0.9998	0.06	0.20
Albiflorin	y = 7.425x − 7.816	31.25–1000	0.9998	0.35	1.17
Paeoniflorin	y = 6.751x − 46.907	31.25–1000	0.9998	0.38	1.27
Benzoic acid	y = 29.191x − 0.899	3.15–100	0.9999	0.02	0.07
Cinnamic acid	y = 22.359x − 2.634	1.0–50	0.9997	0.01	0.03

^a^ y, the peak area of the compound; x, concentration (μg/mL) of the compound. ^b^ LOD, limit of detection, signal-to-noise, S/N = 3. ^c^ LOQ, limit of quantification, S/N = 10.

**Table 2 antioxidants-09-00627-t002:** Content of the five compounds in CPA4-1 as determined by HPLC analysis.

Compounds	Source	Content in CPA4-1 (%)
Gallic acid	Paeoniae Radix	1.1
Albiflorin	Paeoniae Radix	5.3
Paeoniflorin	Paeoniae Radix	8.5
Benzoic acid	Paeoniae Radix	0.4
Cinnamic acid	Cinnamomi Ramulus	0.1

CPA4-1 (Cinnamomi Ramulus:Paeoniae Radix = 1:8).

**Table 3 antioxidants-09-00627-t003:** Inhibitory effect of CPA4-1 on the formation of advanced glycation end product (AGE).

Sample	Conc., μg/mL	Inhibition, %	IC_50_, μg/mL
	2.5	16.64 ± 0.58	
CPA4-1	5	43.06 ± 0.71	6.84 ± 0.08
	10	69.68 ± 1.67	
	55.55 (750 μM)	44.53 ± 2.33	
AG	74.06 (1000 μM)	49.97 ± 0.59	78.28 ± 4.24 (1056.47 ± 57.25 μM)
	111.10 (1500 μM)	56.37 ± 0.67	

CPA4-1 was added into BSA solution and 0.2 M glucose and fructose and incubated for 14 days. AGE-specific fluorescence was analyzed using a spectrofluorometer, as described in the Methods section. IC_50_ (50% inhibition concentration) value was calculated from the dose-inhibition curve (n = 3). Aminoguanidine (AG) was used as a positive control.

**Table 4 antioxidants-09-00627-t004:** Blood and liver parameters in the serum of db/db mice treated with FENO and CPA4-1.

	NOR	DM	FENO	CPA4-1
Blood glucose (mg/dl)	137.6 ± 16.2	447.2 ± 113.7 *	423.5 ± 266.4 ^#^	459.6 ± 146.9
HbA1c (%)	2.66 ± 1.74	11.29 ± 3.81 *	6.32 ± 5.14 ^#^	10.63 ± 1.96

NOR, normal control mice; DM, db/db mice; FENO, db/db mice treated with fenofibrate (100 mg/kg); CPA4-1-100, db/db mice treated with CPA4-1 (100 mg/kg). All data are expressed as mean ± SD. * *p* < 0.05 vs. Nor group, ^#^
*p* < 0.05 vs. DM group. HbA1c, glycated hemoglobin.

## Data Availability

All data generated or analyzed during this study are included in this published article.

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
