# Peer review of "The Herbal Combination CPA4-1 Inhibits Changes in Retinal Capillaries and Reduction of Retinal Occludin in db/db Mice"

_antioxidants, 2020, doi:10.3390/antiox9070627_

Round 1

Reviewer 1 Report

In this paper the effect of CPA4-1 (an herbal combination) on the retina of db/db mice has been examined. The authors conclude that CPA4-1 treatment prevents AGEs accumulation and ameliorated BRB leakage. Comparisons with aminoguanidine and fenofibrate were also performed. This is a well-conducted and interesting research but there are several points that should be addressed:

Introduction:

  • In the first paragraph is stated: “Inhibition of AGE formation has been suggested as a therapeutic target for improving insulin resistance in diabetes with obesity”. Please support this with appropriate references.

Results:

  • Table 3. It is surprising the low levels of HbA1c that present the normoglycemic db/db mice (mean of 2.6%), as well as the powerful capacity of fenofibrate in lowering blood glucose levels. Please review carefully these data and provide any potential explanation.
  • Figure 5 should be improved.

Author Response

Ref.: Ms. ID. antioxidants-861162

Thank you for email on July 7, 2020, informing us of your comments on our manuscript (Manuscript ID: antioxidants-861162, Title: The herbal combination CPA4-1 inhibits changes in retinal capillaries and reduction of retinal occludin in db/db mice). I would now like to submit the revised manuscript to be considered for publication as a communication in the molecules. The manuscript has been modified based on your comments and corrections, our replies to which are included below. In the manuscript, we mark the highlight in the changed part.

Response to Reviewer 1 Comments

In this paper the effect of CPA4-1 (an herbal combination) on the retina of db/db mice has been examined. The authors conclude that CPA4-1 treatment prevents AGEs accumulation and ameliorated BRB leakage. Comparisons with aminoguanidine and fenofibrate were also performed. This is a well-conducted and interesting research but there are several points that should be addressed:

Introduction:

  • In the first paragraph is stated: “Inhibition of AGE formation has been suggested as a therapeutic target for improving insulin resistance in diabetes with obesity”. Please support this with appropriate references. Results: Response: Reference was added in the introduction.
  • Table 3. It is surprising the low levels of HbA1c that present the normoglycemic db/db mice (mean of 2.6%), as well as the powerful capacity of fenofibrate in lowering blood glucose levels. Please review carefully these data and provide any potential explanation. Response: Explanation was added in the discussion part (page 120 line 326~327).
  • Figure 5 should be improved. 

    Response: Figure 5 was changed.

Reviewer 2 Report

The manuscript by SooK Kim et al describes the effect of   CPA4-1 in AGE formation and tested it as a treatment for diabetic retinopathy in a db/db mouse model. The manuscript is very clear and the data support most of their conclusions. The quality of the experiments performed in animals could be dramatically increased if for examples in vivo angiography and OCT were used to assess retinal vasculature/leakage and retinal morphology.

Major:

  1. Once there is no space limitation imposed by this journal, there is no reason to not clearly stated how the experiments were performed in the Material and Methods. The authors should clearly describe in detail how the experiments were done to allow others to replicate the study. Refer to other studies is not sufficient.
  2. I kindly ask the authors to improve the quality of the figures, mainly the lettering size and resolution to increase the readability of the same.
  3. The reference style needs to be adapted to the currently used in Antioxidants.

Minor:

Line 27-28: “db/db mice showed hyperglycaemia, increased Hb1Ac, altered retinal capillaries, and reduced retinal expression of occludin, which is an important tight junction protein.”  The sentence only describes the db/db model are thereafter this information is not needed.

Line 92. “MeOH” should be correctly written or methanol or CH3OH.

Line 128: Sentence should be removed to avoid text publication: “ARPE-19 cells were maintained at 37°C in humidified 95% air/ 5% CO2 atmosphere.”

Line 253: “253 CPA4-1 treatment restored occludin expression in db/db mice.” The sentence should be corrected once there is no significant increase of occludin expression. Similar needs to be performed in line 283.

Figure 6. The full western blot picture for the occluding should be presented, once it is not very clear on the presented crop panel.

Author Response

Ref.: Ms. ID. antioxidants-861162

Thank you for email on July 7, 2020, informing us of your comments on our manuscript (Manuscript ID: antioxidants-861162, Title: The herbal combination CPA4-1 inhibits changes in retinal capillaries and reduction of retinal occludin in db/db mice). I would now like to submit the revised manuscript to be considered for publication as a communication in the molecules. The manuscript has been modified based on your comments and corrections, our replies to which are included below. In the manuscript, we mark the highlight in the changed part.

Response to Reviewer 2 Comments

The manuscript by Young SooK Kim et al describes the effect of CPA4-1 in AGE formation and tested it as a treatment for diabetic retinopathy in a db/db mouse model. The manuscript is very clear and the data support most of their conclusions. The quality of the experiments performed in animals could be dramatically increased if for examples in vivo angiography and OCT were used to assess retinal vasculature/leakage and retinal morphology.

Major:

  1. Once there is no space limitation imposed by this journal, there is no reason to not clearly stated how the experiments were performed in the Material and Methods. The authors should clearly describe in detail how the experiments were done to allow others to replicate the study. Refer to other studies is not sufficient.

Response 1: According to the comments of reviewer material and methods was described in detail (page 3 line 107~111, 1294~135; page 4 line 145~147; page 5 line 181~183).

  1. I kindly ask the authors to improve the quality of the figures, mainly the lettering size and resolution to increase the readability of the same.

Response 2: All figures were enlarged.

  1. The reference style needs to be adapted to the currently used in Antioxidants.

 Response 3: Reference style was changed as Antioxidants.

Minor:

Line 27-28: “db/db mice showed hyperglycaemia, increased Hb1Ac, altered retinal capillaries, and reduced retinal expression of occludin, which is an important tight junction protein.”  The sentence only describes the db/db model are thereafter this information is not needed.

Response: That sentence was removed.

Line 92. “MeOH” should be correctly written or methanol or CH3OH.

Response: “MeOH” was changed to methanol.

Line 128: Sentence should be removed to avoid text publication: “ARPE-19 cells were maintained at 37°C in humidified 95% air/ 5% CO2 atmosphere.”

Response: That sentence was removed.

Line 253: “253 CPA4-1 treatment restored occludin expression in db/db mice.” The sentence should be corrected once there is no significant increase of occludin expression. Similar needs to be performed in line 283.

Response: After finishing the experiment, we divided eyes from each group for several experiment (immunohistochemical staining, western blot, and BRB breakage assay). That result showed the occludin expression using immunostaining and western blot. In immunostaining, DM was clearly increased loss of the tight junction (green arrow) and CPA4-1 group was reduced. For western blot of occlucin, the protein from eye retina were not sufficient to perform statistical repetition

 Figure 6. The full western blot picture for the occludin should be presented, once it is not very clear on the presented crop panel.

Response: Please check the attached PDF file. When performing this experiment as shown in the following figure, the other drug (CMO4-1) was tested simultaneously and one membrane was probed with occludin antibody or β-actin antibody and repeated twice. We needed to crop the membrane just showed the CPA4-1.

I look forward to hearing from you concerning the acceptability of our manuscript in the future.

With best regards.

Sincerely,

Jin Sook Kim, Ph.D.

Chief Research Scientist

Reviewer 3 Report

The manuscript by  Young SooK Kim and colleagues described the impact of herbal blend treatment of diabetic retinopathy, an extremely harsh complication of diabetic.

The authors demonstrated the impact of this blend in vito, in cell culture and in vivo and have proven its effectiveness. However, several questions remain and should be addressed.

Major

  1. The 1:8 ratio of the blend (CPA4-1): No evidence were presented for this selection. Is it better than each compound by itself? Is it synergistic or additive? Is 1:7 still effective? The author should demonstrate the superiority of the blend or the importance of Ramulus, the minor compound.
  2. Active compounds presented in Figure 1. It is not clear if the compounds are the active molecules or just markers for standardizing the extraction. Has a synthetic blend [in the same ratio] have been evaluated (in vito or in cells) to determine that their present is sufficient or necessary for the effect?

Minor:

  • P2 line 59: “Naturally occurring phytochemicals have been found to be relatively non-toxic as compared to synthetic AGEs inhibitor”. Please remove statements, as this is not correct. Natural compounds can be more toxic than synthetic sometimes.   
  • Figure 1: please specify LOD,LOQ, and linearity of all markers.
  • Line 194 CPA4-1 was 11.44-fold more potent and not effective.
  • Line 201 , the impressive impact does not look as dose-dependent (2A). Please change.

Author Response

Ref.: Ms. ID. antioxidants-861162

Thank you for email on July 7, 2020, informing us of your comments on our manuscript (Manuscript ID: antioxidants-861162, Title: The herbal combination CPA4-1 inhibits changes in retinal capillaries and reduction of retinal occludin in db/db mice). I would now like to submit the revised manuscript to be considered for publication as a communication in the molecules. The manuscript has been modified based on your comments and corrections, our replies to which are included below. In the manuscript, we mark the highlight in the changed part.

Response to Reviewer 3 Comments

Comments and Suggestions for Authors

The manuscript by Young SooK Kim and colleagues described the impact of herbal blend treatment of diabetic retinopathy, an extremely harsh complication of diabetic.

The authors demonstrated the impact of this blend in vitro, in cell culture and in vivo and have proven its effectiveness. However, several questions remain and should be addressed.

Major

  1. The 1:8 ratio of the blend (CPA4-1): No evidence were presented for this selection. Is it better than each compound by itself? Response: As shown in the table below, previous study we performed the different ratio of two herbs on inhibition assay of AGEs formation. We mentioned the results of inhibition on AGE formation in the introduction (page 2 line 74~76) and discussion (page 12 line 308~309).

Sample

Conc., μg/mL

Inhibition, %

IC50, μg/mL

2.5

12.22±0.78

25.66±0.65

50.14±1.06

9.94±0.20

CPA1-1(1:1)

5

10

2.5

10.60±0.77

CPA2-1(1:2)

5

30.09±1.59

7.54±0.18

10

69.25±1.63

2.5

12.33±0.74

CPA3-1(1:4)

5

25.83±0.66

8.16±0.23

10

63.62±2.27

2.5

16.64±0.58

CPA4-1(1:8)

5

43.06±0.71

6.84±0.08

10

69.68±1.67

55.55 (750 μM)

44.53±2.33

AG

74.06 (1000 μM)

49.97±0.59

78.28±4.24 (1056.47±57.25 μM)

111.10 (1500 μM)

56.37±0.67

Is it synergistic or additive? Response: It is not synergistic effect. There is no herb as a positive control. We used aminoguanidine as a control.

Is 1:7 still effective? Response: We did not try 1:7 ratios. However, we think it will work, referring to the table above.

The author should demonstrate the superiority of the blend or the importance of Ramulus, the minor compound.

Response: This paper suggest the effect of CPA4-1 (Cinnamomi Ramulus : Paeoniae Radix=1:8) on diabetic retinopathy. Previous study that Cinnamomi Ramulus has traditionally been used for its anti-inflammatory, antioxidant, and neuroinflammatory effects. Single herb extract was not focused on this paper.

  1. Active compounds presented in Figure 1. It is not clear if the compounds are the active molecules or just markers for standardizing the extraction. Has a synthetic blend [in the same ratio] have been evaluated (in vitro or in cells) to determine that their present is sufficient or necessary for the effect?

Response: As reviewer’s comment, we changed the active to marker compounds (page 2 line 67 and 72).

Minor:

  • P2 line 59: “Naturally occurring phytochemicals have been found to be relatively non-toxic as compared to synthetic AGEs inhibitor”. Please remove statements, as this is not correct. Natural compounds can be more toxic than synthetic sometimes.   
  • Response: As reviewer’s comment that sentence was removed. 

  • Figure 1: please specify LOD, LOQ, and linearity of all markers. 
  • Response : For regression equation, linearity, LOD and LOG for 5 marker compounds, table 1 was added (page 5 line 202).

  • Line 194 CPA4-1 was 11.44-fold more potent and not effective. 
  • Response: As reviewer’s comment, the “effective” was changed (page 6 line 213).

  • Line 201, the impressive impact does not look as dose-dependent (2A). Please change.  Sincerely,Chief Research ScientistEncl.
  • Response: As reviewer’s comment, the sentence was changed (page 7 line 222).

I look forward to hearing from you concerning the acceptability of our manuscript in the future.

With best regards.

Jin Sook Kim Ph.D.

Chief Research Scientist

Encl.

 cc. Co-authors

Round 2

Reviewer 1 Report

No further comments.

Reviewer 3 Report

n/a